# Backscattering-free edge states below all bands in two-dimensional auxetic media

Wenting Cheng [1,4], Kai Qian [2,4], Nan Cheng [1,4], Nicholas Boechler [2,3] ✉, Xiaoming Mao [1] ✉ & Kai Sun [1] ✉

Unidirectional and backscattering-free propagation of sound waves is of fundamental interest in physics and highly sought-after in engineering. Current strategies utilize topologically protected chiral edge modes in bandgaps, or complex mechanisms involving active constituents or nonlinearity. Here we propose passive, linear, one-way edge states based on spin-momentum locking of Rayleigh waves in two-dimensional media in the limit of vanishing bulk to shear modulus ratio, which provides perfect unidirectional and backscattering-free edge propagation that is immune to any edge roughness and has no limitation on its frequency (instead of residing in gaps between bulk bands). We further show that such modes are characterized by a topological winding number that protects the linear momentum of the wave along the edge. These passive and backscattering-free edge waves have the potential to enable phononic devices in the form of lattices or continua that work in previously inaccessible frequency ranges.

One-way transport of elastic waves is of fundamental interest to a broad range of applications such as vibration isolation, rectification, and communication[1]. While such one-way transport is prohibited by the law of reciprocity in linear systems with time-reversal symmetry (TRS)[2,3], extensive research in the past decade has led to promising designs based on active driving[4,5], nonlinearity[6,7], or topological states[8–13]. However, each of these designs comes with fundamental limitations. Active systems require constant energy consumption, nonlinear mechanisms induce signal distortion and are dependent on wave amplitudes, and topological edge states are limited to narrow gaps between bulk bands.

A related but even more sought-after, property is "backscattering-free" wave propagation, wherein the one-way transport not only happens in media with perfect homogeneity and straight edges but is also retained in the presence of impurities and edge roughness. The canonical example of this behavior is the quantum Hall effect (QHE), where, under broken time-reversal symmetry, bandgap topological edge states only exist in one propagation direction[14,15]. The strong topological protection from the Chern number, in this case, makes it immune to impurity and edge roughness. Mechanical analogs of the

QHE have been proposed recently, where active elements were introduced to break TRS[8,9]. However, unlike the case of electrons, mechanical QHE analogs require active driving. Passive topological mechanical systems have also been proposed, based on the quantum spin and valley Hall effects, where edge waves carrying opposite polarizations travel in opposite directions[10,16–20]. However, as demonstrated recently, these designs require precise engineering of the edges and interfaces that preserve their underlying symmetries and are thus not generally backscattering-free[21,22].

Here, we report the backscattering-free edge waves in passive linear auxetic elastic media in two dimensions (2D), where the formal nature of the wave functions in the limit of a vanishing ratio between the bulk and the shear moduli ($B/\mu \to 0$) provides a topological protection for these waves to propagate in a unidirectional fashion under any edge roughness. Further, these modes can exist over an infinite frequency range, in contrast to the requirement of prior topologically protected modes to live in finite frequency bandgaps. Our design leverages the spin-momentum locking of elastic waves[23–31] at the surface of 2D materials. It is important to note that the spin here is the true physical spin of the elastic waves in continuous media, and the

[1]Department of Physics, University of Michigan, Ann Arbor, MI, USA. [2]Department of Mechanical and Aerospace Engineering, University of California San Diego, La Jolla, CA, USA. [3]Program in Materials Science and Engineering, University of California San Diego, La Jolla, CA, USA. [4]These authors contributed equally: Wenting Cheng, Kai Qian, Nan Cheng. ✉e-mail: nboechler@ucsd.edu; maox@umich.edu; sunkai@umich.edu

underlying principles proposed in this study, based on conformal symmetry, are fundamentally different from those in the existing literature. This mechanism exhibits exceptional robustness against edge roughness, does not require specific microstructures or symmetries, and thus opens pathways for achieving backscattering-free one-way waves in practice. We experimentally verify this effect using auxetic Maxwell lattices (which enable $B/\mu \to 0$). We formulate an analytic theory to show that this locking is topologically protected by a real space winding number of the wave function and thus robust against any edge roughness, endowing true "backscattering-free" one-way transport to these waves, and demonstrate this effect computationally. Because of the aforementioned infinite bandwidth, wherein our modes exist below all bulk bands (in the sense of having lower wave speed), we also highlight that these modes can occur at arbitrarily low frequencies. This enables our design to reach not only the backscattering-free limit but also the damping-free limit, which was previously inaccessible for one-way topological mechanical waves. This combination of a passive, infinitely broadband, generally backscattering-free, and damping-free material has significant potential importance, as it may enable long-distance and low-energy mechanical communication and computing.

## Results

### One-way transport and spin-momentum locking of Rayleigh waves

How to excite a wave that only travels in $+\mathbf{k}$ and not $-\mathbf{k}$? In principle, this can be achieved by applying forces on the medium that exactly follow the $+\mathbf{k}$ mode in space-time. A perhaps more interesting question, however, is how to create one-way propagation when one can only apply a periodic force $\mathbf{F}$ at one point $\mathbf{r}$ on the edge of the medium. For simplicity, we first consider the case where the material is a semi-infinite plane and then discuss general geometries next. This point source will generate a response in all modes at (angular) frequency $\omega$, and the amplitude of each mode $\boldsymbol{\psi}_\alpha$ will be proportional to $\boldsymbol{\psi}_\alpha(\mathbf{r})^* \cdot \mathbf{F}$, where $^*$ denotes complex conjugation (see Supplementary Materials (SM) for details). Given that $\mathbf{r}$ is on the edge, this overlap is much greater for edge-localized modes than bulk waves. Furthermore, these edge modes appear in pairs of $\boldsymbol{\psi}_\mathbf{k}$ and $\boldsymbol{\psi}_{-\mathbf{k}}$, where $\mathbf{k}$ is the wavevector along the edge, and a generic point source $\mathbf{F}$ excites both $\boldsymbol{\psi}_\mathbf{k}$ and $\boldsymbol{\psi}_{-\mathbf{k}}$.

To approach "one-way transport" in the $+\mathbf{k}$ direction, one needs to maximize the ratio of the amplitudes of $\boldsymbol{\psi}_\mathbf{k}$ and $\boldsymbol{\psi}_{-\mathbf{k}}$. This can be achieved by choosing $\mathbf{F} = \boldsymbol{\psi}_\mathbf{k}(\mathbf{r})$. A metric for asymmetric edge transport can then be defined as

$$\left| \frac{C_{-\mathbf{k}}}{C_\mathbf{k}} \right| = \left| \frac{\boldsymbol{\psi}_{-\mathbf{k}}(\mathbf{r})^* \cdot \mathbf{F}}{\boldsymbol{\psi}_\mathbf{k}(\mathbf{r})^* \cdot \mathbf{F}} \right| = \left| \frac{\boldsymbol{\psi}_{-\mathbf{k}}(\mathbf{r})^* \cdot \boldsymbol{\psi}_\mathbf{k}(\mathbf{r})}{\boldsymbol{\psi}_\mathbf{k}(\mathbf{r})^* \cdot \boldsymbol{\psi}_\mathbf{k}(\mathbf{r})} \right|, \qquad (1)$$

which vanishes in the limit of one-way transport. This ratio not only describes how external forces at the boundary point $\mathbf{r}$ excite left (L) vs. right (R) propagating waves, but also describes impurity scattering as the wave propagates in the bulk close to the edge, where impurities create new sources $\mathbf{F}$. One example where $|C_{-\mathbf{k}}/C_\mathbf{k}| = 0$ is the quantum spin Hall effect, where spin-polarized topological edge states lead to orthogonal states for $\mathbf{k}$ and $-\mathbf{k}$.

Here, we consider instead Rayleigh waves traveling on the edge ($y = 0$) of a 2D ($x$-$y$) semi-infinite isotropic continuum plane, which take the form at the edge

$$\boldsymbol{\psi}_{\pm\mathbf{k}}(x, 0, t) = e^{i(\pm kx - \omega t)} \begin{bmatrix} \frac{1}{2}\xi\sqrt{1 + \frac{\mu}{B}} \\ \pm \frac{1}{4}i\xi(2 - \xi^2)\sqrt{\frac{1 + (\mu/B)}{1 - \xi^2}} \end{bmatrix}, \qquad (2)$$

where $\xi = \xi(B/\mu)$ is the ratio of Rayleigh to transverse wave speeds and a function of the ratio of the 2D bulk modulus $B$ and shear modulus $\mu$ (see Supplementary Materials for details). This pair of waves are linear

combinations of longitudinal and transverse waves, satisfying stress-free boundary conditions at $y = 0$, and their longitudinal and transverse components decay into the bulk with different decay lengths[32]. These Rayleigh waves have speeds equal to or less than the bulk waves, and their dispersion is linear in semi-infinite continua, resulting in the aforementioned infinite bandwidth (and residence below all bands). The 2D moduli are related to those in 3D (denoted by the 3D subscript) by considering a 3D plate in a plane stress state, such that $B = \frac{3B_{3D}\mu_{3D}}{B_{3D} + \frac{4}{3}\mu_{3D}}, \mu = \mu_{3D}$.

Interestingly, at the limit of $B/\mu \to 0$, (equivalently $\xi \to 0$), this pair of waves at the surface become

$$\boldsymbol{\psi}_{\pm\mathbf{k}}(x, 0, t) = \frac{1}{\sqrt{2}} e^{i(\pm kx - \omega t)} \begin{bmatrix} 1 \\ \pm i \end{bmatrix}, \qquad (3)$$

which are circularly polarized. In this case, $|C_{-\mathbf{k}}/C_\mathbf{k}| = 0$, leading to 100% one-way transport in the limit of $B/\mu \to 0$. The general case of $B/\mu > 0$ leads to ellipsoidal polarization and $|C_{-\mathbf{k}}/C_\mathbf{k}| > 0$. In the limit of $B/\mu \to \infty$, $|C_{-\mathbf{k}}/C_\mathbf{k}| \to 0.4203 < 1$, meaning that the propagation along $\pm\mathbf{k}$ is still biased (Fig. 1a, c, with simulation described in Methods). It is also worth mentioning that in the limit of $B/\mu \to 0$, the Rayleigh wave becomes more localized to the surface compared to the wavelength as the decay factor grows larger, which also minimizes the excitation of the bulk waves from the point source on the edge (see Supplementary Materials Section SII).

The circular polarization of the Rayleigh waves in the limit of $B/\mu \to 0$ suggests that a spin-momentum locking mechanism can be defined for this one-way transport. Indeed, similar to the spin angular momentum of photons, where photons of $\pm 1$ spins correspond to quanta of R and L polarized electromagnetic waves, a spin angular momentum can be defined for phonons[33–35]. In the Supplementary Materials, we show that the rotational symmetry of the media leads to conserved angular momentum density, following Noether's theorem[33–36], and this angular momentum $\mathbf{j} = \mathbf{l} + \mathbf{s}$ can be expressed as a sum of an orbital ($\mathbf{l}$) and spin ($\mathbf{s}$) angular momentum density of the wave, where

$$\mathbf{l} = \mathbf{r} \times \mathbf{p}, \quad \mathbf{s} = \rho \, \mathbf{u} \times (\partial_t \mathbf{u}), \qquad (4)$$

$\rho$ is the mass density, $\mathbf{r}$ is the position, and $\mathbf{p} = -\rho \, \partial_t u_i \mathbf{u}_i$ is the density of linear momentum. Following this, the Rayleigh waves carry physical spin angular momentum pointing along $\hat{\mathbf{z}}$ (normal to the 2D plane), and under standard canonical quantization[37], the spins of these phonons take values of $\pm 1$. It is worth noting that this spin characterizes local circular motions of material points in the medium, and as such, these modes carry spin angular momentum. Further, we note that this should not be confused with rigid-body rotational fields defined in theories such as Cosserat elasticity[38].

What about 3D Rayleigh waves? A similar calculation can be done, giving a pair of Rayleigh waves for any propagation direction on the 2D surface of a semi-infinite 3D solid (see Supplementary Materials for details)[23,39]. However, the asymmetric edge transport ratio never reaches zero ($|C_{-\mathbf{k}}/C_\mathbf{k}| > 0.0509$, Fig. 1b, d). Thus, perfect one-way transport, in the manner defined herein using Rayleigh waves, is only achieved for 2D solids. More importantly, 3D Rayleigh waves are far from "perfect" in terms of spin-momentum locking, even in the $B_{3D}/\mu \to 0$ limit. This is because, in a 3D solid, although the Rayleigh-wave mode shape is close to circular polarization at the surface layer, as one moves into the bulk, the Rayleigh-wave mode shape quickly becomes elliptical[23] (Fig. 1f), i.e., $|C_{-\mathbf{k}}/C_\mathbf{k}| > 0$ in the bulk, and thus any impurity would result in backscattering. In contrast, 2D Rayleigh waves have $|C_{-\mathbf{k}}/C_\mathbf{k}| = 0$ at all depths (Fig. 1e), making them immune to backscattering.

Here, we note two additional important points. The first point is that the 2D Rayleigh wave speed $c_R$ changes as $B/\mu \to 0$, as can be seen

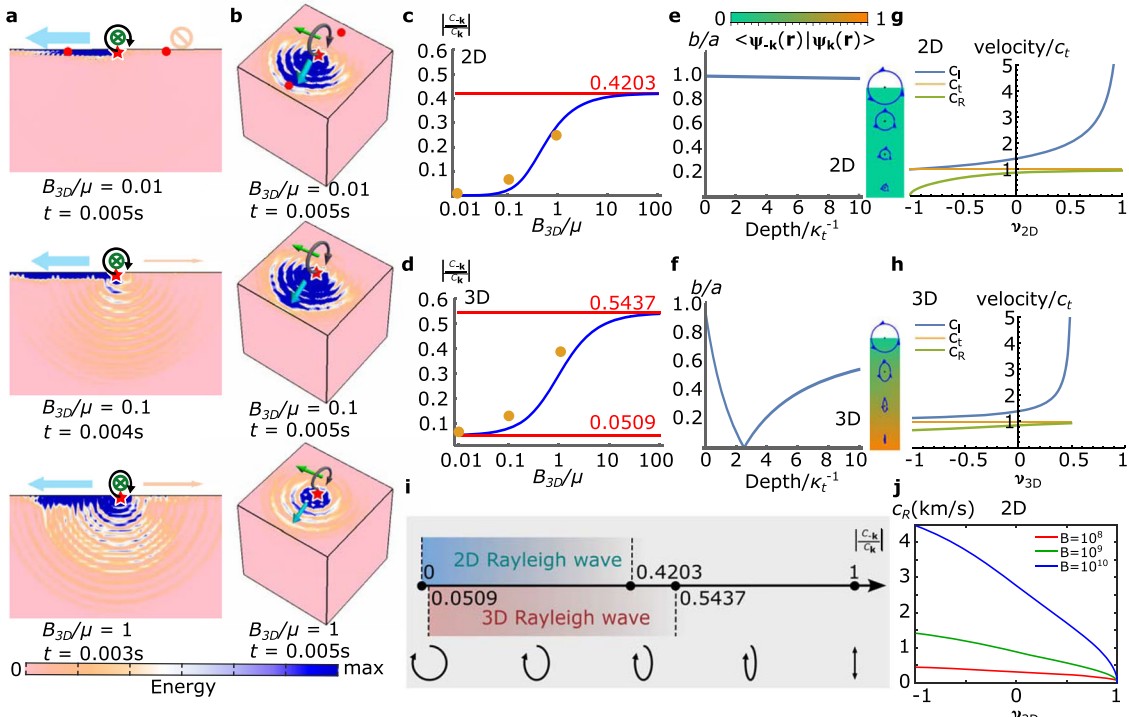

**Fig. 1 | One-way transport and spin-momentum locking of Rayleigh waves.**
Simulated spatial energy distribution for 2D (**a**) and 3D (**b**) continua with different $B_{3D}/\mu$ values, using a prescribed particle displacement at boundary point **r** following $\psi_k(\mathbf{r})$. Green arrows denote the spin direction (parallel to the surface) and blue arrows denote the strong propagation direction. The relative displacement magnitude of the red points marked in (**a** and **b**) is evaluated to measure $|C_{-k}/C_k|$, shown as yellow points in (**c** and **d**). The shear modulus ($\mu$) is kept constant at 0.01 GPa throughout all the simulations, and the bulk modulus ($B_{3D}$) varied. **c**, **d** Dependence of the one-way propagation metric on the elastic moduli ratio in 2D (**c**) and 3D (**d**). The blue curves are analytic results, and the red lines are bounds. **e**, **f** The dependence of polarization of the Rayleigh wave at various depths when $B_{3D}/\mu = 0$ (normalized using the decay length $\kappa_t$ of the transverse polarized

component of the edge mode), depicted as the ratio of the short to long axes ($b/a$) of the ellipse of polarization (analytic). The inset illustrates the polarization (blue ellipses) and $\psi_{-k}(\mathbf{r})^* \cdot \psi_k(\mathbf{r})$ (color) of the Rayleigh waves at different depths for 2D (**e**) and 3D (**f**). Sizes of the ellipses are proportional to the amplitude of the Rayleigh waves at the ellipses' center. **g**, **h** Dependence of the longitudinal ($c_l$), transverse ($c_t$), and Rayleigh ($c_R$) wave speeds (normalized by $c_t$) on the Poisson's ratio ($\nu$) in 2D (**g**) and 3D (**h**) (analytic). **i** A summary of the one-way transport metric in 2D and 3D media. **j** Dependence of Rayleigh wave speeds ($c_R$) on Poisson's ratio ($\nu$) in 2D continua at $B = 10^8, 10^9, 10^{10}$ Pa (analytic), indicating that $c_R$ approaches a finite value ($\sqrt{2B/\rho}$, and $\rho = 1000$ kg/m$^3$) as $\nu \to -1$ (equivalently $B/\mu \to 0$) when $B$ is kept constant, and $\mu$ is increased (see Supplementary Materials for details).

in Fig. 1g, h, j. We note that this is equivalent to the Poisson's ratio approaching $-1$, where the Poisson's ratio in 2D and 3D are expressed as $\nu_{2D} = \frac{B-\mu}{B+\mu}$ and $\nu_{3D} = \frac{3B_{3D}-2\mu}{2(3B_{3D}+\mu)}$, respectively. As shown in the Supplementary Materials, as $B/\mu \to 0$, $c_R \to \sqrt{2B/\rho}$. Thus, achieving $B/\mu \to 0$ by decreasing $B$ results in a vanishing $c_R$, whereas achieving it by increasing $\mu$ results in a finite $c_R$. Along these lines, in the limit of $B/\mu \to 0$, if $\mu$ is held finite, then the bulk wave speeds approach $c_t = c_l = \sqrt{\mu/\rho}$. Conversely, if $B$ is held finite and $\mu \to \infty$, then both bulk wave speeds become infinite, while the Rayleigh wave speed remains finite. The second point is that (1) addresses one-way transport, giving the conditions under which perfect one-way transport can be achieved upon point excitation as $B/\mu \to 0$. In the subsequent section, we demonstrate that $B/\mu \to 0$ is also the condition for these waves to achieve the more challenging feature of backscattering-free transport.

**Backscattering-free one-way transport in arbitrary geometries**
The Rayleigh wave formulation discussed above offers a concise picture to characterize how spin-momentum locking and one-way transport arise in these edge waves in the simple geometry of semi-infinite media with a straight edge. We find that the observed one-way transport is also robust in 2D materials with very irregular geometries, as shown in Fig. 2 for finite objects, and in the SM for semi-infinite media with surface scatterers.

To reveal the physical origin of this phenomenon, we propose a formulation based on mapping of zero modes in the $B/\mu \to 0$ limit in 2D to complex analytic functions and perturbatively extending them to

low-frequency edge waves at small $B/\mu$. By doing this, we show that such spin-momentum locking is a general phenomenon in 2D $B/\mu \ll 1$ materials of any shape. Further, we show that these edge waves are characterized by a topological winding number that plays the role of discrete angular momentum eigenvalues despite the arbitrary geometry and protects the linear momentum of the wave along the edge.

We first use the complex plane to rewrite the 2D elasticity problem, by defining coordinate $z = x + iy$ and displacement field $\psi = u_x + iu_y$. In the limit of $B/\mu \to 0$, the wave equation and boundary conditions can then be written as (see Supplementary Materials for details)

$$\rho \frac{\partial^2 \psi(z, z^*, t)}{\partial t^2} = 4\mu \frac{\partial^2 \psi(z, z^*, t)}{\partial z \partial z^*}, \quad z \in \Omega,$$
$$\frac{\partial \psi(z, z^*, t)}{\partial z^*} = 0, \quad z \in \partial\Omega,$$

$$(5)$$

where $\Omega$ and $\partial\Omega$ represent the bulk and edge of the system, respectively. Solutions of (5) can be obtained via the separation of variables $\psi_\pm(z, z^*, t) = \phi(z, z^*) e^{\pm i\omega t}$. Because $e^{+i\omega t}$ rotates the displacement vector counterclockwise and $e^{-i\omega t}$ rotates it clockwise over time, these two solutions can be said to correspond to spin up and down states. This equation has $\omega = 0$ solutions $\phi_0$ given by $\partial\phi_0/\partial z^* = 0$, i.e., arbitrary bounded analytic functions $\phi_0(z)$. These solutions have been characterized as static zero modes in ref. 40, and they can be derived from the fact that zero modes in 2D materials at $B/\mu = 0$ are conformal transformations, which map to analytic functions[40,41].

Here, we are interested in the dynamics of waves at small but nonzero $B/\mu$, which can be solved perturbatively from the zero modes. The same as the perturbation theory for the Schrödinger equation[42] to the leading order, small perturbations only modify the frequencies (eigenvalues), while wavefunctions remain analytic functions. Thus, at small $B/\mu$, these zero modes acquire finite frequencies $\omega \propto \sqrt{B/\mu}\,\omega_0$ (where $\omega_0$ is the first bulk mode frequency) and take the form

$$\psi_\pm(z, z^*, t) = \phi_0(z)\, e^{\pm i\omega t}. \qquad (6)$$

How do these waves propagate in space and time? The irregular geometry we consider here exhibits no translational or rotational symmetries, so linear momentum $\mathbf{k}$ and orbital angular momentum $\mathbf{l}$ are no longer conserved quantities. Instead, a winding number can be defined by applying the argument principle on the analytic functions $\phi_0$, which plays a role analogous to orbital angular momentum eigenvalues in quantum mechanics, such that

$$\frac{1}{2\pi i}\int_{\partial\Omega} d\ln(\phi_0(z)) = \frac{1}{2\pi i}\int_{\partial\Omega} \frac{\phi_0'(z)}{\phi_0(z)} dz = N_0 - N_\infty, \qquad (7)$$

where $N_0$ and $N_\infty$ are the numbers of zeros and poles, respectively, of $\phi(z)$ inside the bounded domain $\Omega$. The function $\phi_0(z)$ being bounded in $\Omega$ implies $N_\infty = 0$. On the other hand, edge localized modes $\phi_0(z)$ must have zeros in $\Omega$ (Rouché's theorem, as detailed in Supplementary Materials), so $N_0 \geq 1$. This indicates that all zero modes $\phi_0$ wind its phase by at least one cycle of $2\pi$ as one goes along the boundary counter-clockwise. For a finite 2D material at $B/\mu = 0$, there are infinitely many such zero modes, and they each exhibit a winding number $N_0$, which is also the number of points in the bulk where the amplitude of the zero mode reaches zero, counting multiplicity, as is shown in Fig. 3b.

We note that this topological winding number is distinct from the Kane-Lubensky topological winding number defined in ref. 43. These two types of winding numbers describe completely different phenomena and they are not interchangeable. Kane-Lubensky winding numbers, along with other winding numbers based on momentum space winding of wave functions, describe how the bulk states govern the edge modes through the bulk-edge correspondence. In contrast, the winding number defined here is winding in real space and is based on conformal symmetry in the $B/\mu \to 0$ limit, which comes from both the bulk equation of motion and the free boundary conditions. This phenomenon is fundamentally distinct and cannot be

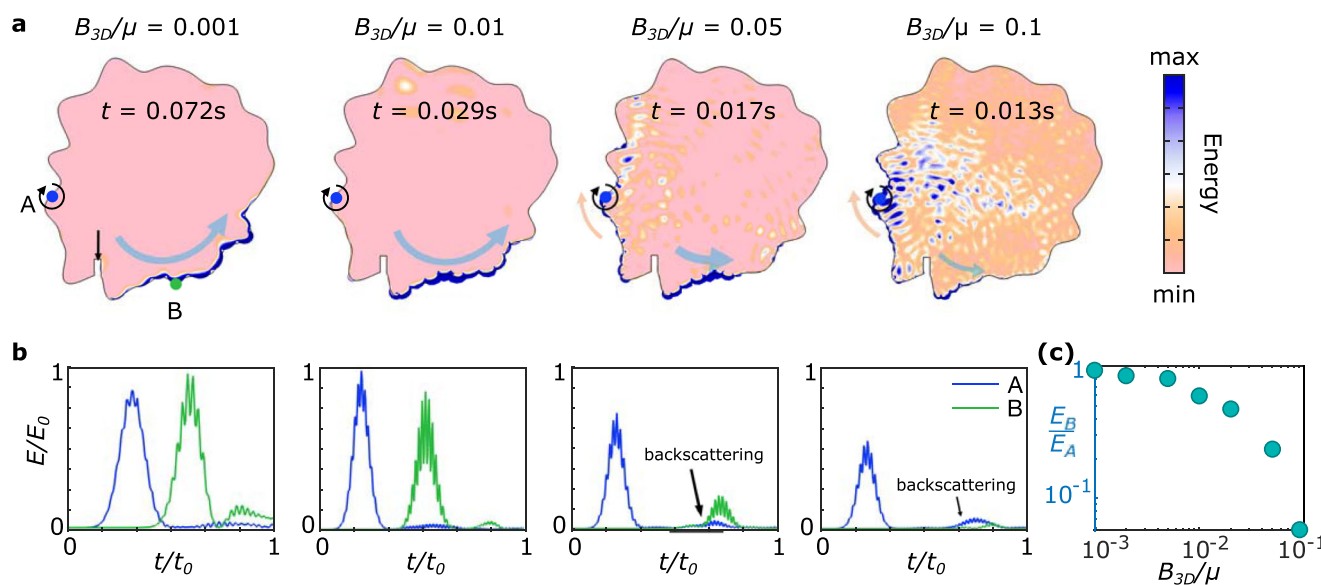

**b** Energy density of points A (actuation point), B (each marked in **a**) as functions of time at these $B_{3D}/\mu$ parameters. Here, $t_0$ and $E_0$ are defined as $t_0 = L/c_R$, $E_0 = \rho c_R^2$, where $L$ is the distance between A and B along the edge, and $c_R$ is the Rayleigh wave speed. **c** The energy density ratio $E_B/E_A$ between points B and A (cumulative over time), demonstrating increasing backscattering immunity as $B_{3D}/\mu \to 0$.

**Fig. 2 | Backscattering-free propagation of one-way edge waves in arbitrary geometries. a** Simulated energy distribution for 2D continua with different values of $B_{3D}/\mu$: 0.001, 0.01, 0.05, and 0.1 (at different times). Clockwise actuation (Gaussian modulated sine wave packet) excites right-going edge waves. To characterize the backscattering, a sharp defect is created at the boundary (black arrow).

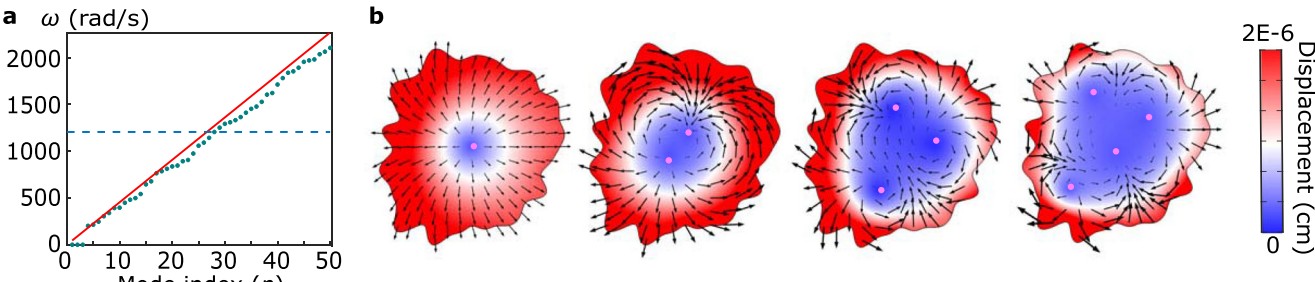

**Fig. 3 | Eigenfrequencies and phase-winding of one-way edge waves in arbitrary geometries. a** Computed (green dots) and analytic (red line, following (10)) eigenfrequencies of a 2D media of irregular shape plotted against the mode index $n$ at $B_{3D}/\mu = 0.01$. Bulk modes start to appear above the blue dashed line, which are interspersed with edge modes at higher $N_0$. **b** Profiles of modes $n = 4, 6, 8, 10$

(corresponding to winding numbers $N_0 = 1, 2, 3, 4$) as examples of waves described by (6). Pink dots mark $\phi = 0$ points in the bulk, the number of which equals the winding number $N_0$(7). The black arrows represent the displacement vectors, which rotate $N_0$ times around the boundary and vanish at $N_0$ points in the bulk.

characterized by conventional winding numbers and bulk-edge correspondence.

The spatial winding of $\phi_0$ and the temporal evolution defined in (6) immediately lead to spin-momentum locking at small $B/\mu$: constant phase points of $\psi_+(z, t) = \phi_0(z) e^{+i\omega t}$ move clockwise and constant phase points of $\psi_-(z, t) = \phi_0(z) e^{-i\omega t}$ move counterclockwise in time. Therefore, this winding number is the extension of discrete angular momentum eigenvalues from a perfect circular geometry to arbitrary geometries in the limit of $B/\mu \to 0$.

It is worth noting that such an extension is not generally defined for any 2D material, where wave eigenstates are usually affected by irregular shapes of the boundary and do not exhibit a particular propagation direction or momentum (see examples in Fig. 2).

It is the $B/\mu \to 0$ limit that permits such a clean definition of an orbital angular momentum. This phase winding of edge modes is not only positive when integrated over the whole edge of the disk, but also positive point-wise along the edge, as shown in the following. Letting $\phi_0(z) = |\phi_0(z)|e^{i\theta(z)}$, the analyticity of $\phi_0(z)$ implies

$$(\boldsymbol{\tau} \cdot \boldsymbol{\nabla})\theta(s) = |\phi_0(s)|^{-1}(\boldsymbol{n} \cdot \boldsymbol{\nabla})|\phi_0(s)|, \tag{8}$$

where $\boldsymbol{n}$ is the outpointing normal vector at the boundary point $s$, and $\boldsymbol{\tau}$ is the counterclockwise oriented tangent vector at $s$. Because $\phi_0(z)$ is an edge mode, in the material, $(\boldsymbol{n} \cdot \boldsymbol{\nabla})|\phi_0(s)| > 0$, leading to the coherent increase of $\theta$ as one moves counterclockwise along the edge. This, together with the $e^{\pm i\omega t}$ factor leads to unidirectional transport.

The two waves $\psi_\pm(z, t)$ are also orthogonal, satisfying the one-way transport criterion $|C_{-\mathbf{k}}/C_{\mathbf{k}}| = 0$ defined in (1). This can be seen by returning from the complex plane notation to the conventional wave notation with $x, y$ components, where

$$\phi_0(z) e^{\pm i\omega t} \to \mathrm{Re}\left\{\phi_0(x + iy)e^{i\omega t}(1, \pm i)^T\right\}, \tag{9}$$

which are circularly polarized, with L and R propagating waves orthogonal to one another. This orthogonality is immune to shape irregularities of the boundary, as it stems from exact zero modes in the $B/\mu \to 0$ limit. Thus, these circularly polarized waves approach backscattering-free as $B/\mu \to 0$ for any geometry (Fig. 2). Note that although these waves adiabatically evolve into Rayleigh waves when the edge becomes straight, the immunity to shape irregularity and backscattering is a distinct property in the $B/\mu \to 0$ limit and not common for general Rayleigh waves. In addition, this backscattering-free one-way transport is independent of the wavelength. Under the condition of a fixed wavelength equal to the defect size, the immunity to backscattering increases as $B/\mu \to 0$ (see Supplementary Figs. S3 and S4).

The spectrum of these edge modes also resembles waves characterized by discrete angular momentum. For 2D finite materials with shapes of no particular symmetry, to leading order in $B/\mu$ (see Supplementary Materials Section SVI for details), these modes have frequencies

$$\omega_n \simeq \frac{2\pi N_0(n) v_{\mathrm{phase}}}{L} \simeq \frac{\pi n v_{\mathrm{phase}}}{L} \tag{10}$$

where $n$ labels the modes, $L$ is the perimeter of the bounded domain $\Omega$ and $v_{\mathrm{phase}} = \xi(B/\mu)\sqrt{\mu/\rho}$ is the phase velocity of Rayleigh waves of the given material. The winding number is related to the mode index by $N_0(n) = \lceil (n-3)/2 \rceil$ where $\lceil \cdot \rceil$ is the ceiling function (noting that each zero mode at $B = 0$ generates a pair of edge modes at $B > 0$, and there are three rigid body modes that remain at $\omega = 0$), which reduces to $N_0(n) \simeq n/2$ at large $n$, leading to the linear dependence of frequency on $n$ shown in Fig. 3a. For a given disk, these modes may extend to infinite $N_0$, although we do not have an analytic proof for irregular disks yet. However, the first bulk mode appears at a finite frequency

(dashed line in Fig. 3a), at the scale of $\sqrt{\mu/\rho}/L$, noting that the longitudinal and transverse wavespeeds converge as $B/\mu \to 0$ (see Fig. 1g). Above this line, the edge modes coexist with the bulk modes. Physically, we have a group of edge waves, the phase of which winds around the irregular edge of the disk by multiples of $2\pi$, and more importantly, the phase dynamically rotates as a function of time, moving the wave around the edge.

The topological protection of the linear momentum of the wave around the edge by the real space winding number (7) can be shown as follows. We write the wave function along the edge as $\phi(s, t) = |\phi_0(s)|e^{i[\theta(s) \pm \omega t]}$ where $s$ is the (complex) coordinate along the edge. The displacement field $(u_x, u_y)$ are again the real and imaginary part of $\phi$. Following the formula for the linear momentum of the wave, $g_i = -\rho \partial_t u_j \partial_i u_j$ (see Supplementary Materials for details), we have

$$g_{\boldsymbol{\tau}}(s, t) = \mp \rho \omega |\phi_0(s)|^2 (\boldsymbol{\tau} \cdot \boldsymbol{\nabla})\theta(s), \tag{11}$$

the sign of which remains the same along the edge (as per Eq. (8)). Given that the sign of $(\boldsymbol{\tau} \cdot \boldsymbol{\nabla})\theta(s)$ remains the same around the edge (8), and the integral of the phase winding around the edge $\int_{\partial\Omega}(\boldsymbol{\tau} \cdot \boldsymbol{\nabla})\theta(s) ds = 2\pi N_0 > 0$ (Eq. (7)), the linear momentum is topologically protected by the winding number.

We revisit here the property that the one-way transport here is below all bands. In the case of the Rayleigh wave solution discussed in the previous section, at each wave vector $k$, the Rayleigh wave is at a lower frequency than all of the bulk modes. In the case of the finite disks discussed in this section, the lowest nonzero bulk mode has a frequency $\omega_0$, and all modes at $0 < \omega < \omega_0$ (Fig. 3) are solely backscattering-free edge modes (noting that backscattering-free edge modes also exist $> \omega_0$, but they are interspersed with the bulk modes). This feature permits backscattering-free one-way edge waves in essentially arbitrarily low frequencies, opening diverse application avenues.

To further elucidate the backscattering-free feature of these waves, we consider an L (or R) polarized wave propagating on an edge and encountering some local roughness or impurities, which in general can be represented by a Hamiltonian $H_I$, which is in addition to the original Hamiltonian operator $H = 4\mu \frac{\partial^2}{\partial z \partial z^*}$ (see Eq. (5) and Supplementary Materials). Its effect can be characterized following standard perturbation theory, where the scattering $S_I$ from L to R (or the opposite) wave eigenstates is controlled by the off-diagonal component of $H_I$ in the L and R bases, such that

$$S_I = |\langle L|H_I|R \rangle|^2. \tag{12}$$

Because L and R waves carry opposite spins, only perturbations that flip the phonon spin can cause backscattering. For instance, impurities with large $B$ cause backscattering as they invalidate the conformal description. Formally, this is similar to the case of QSHE, but we highlight here that the phonon spins are real physical spins so no sublattice symmetry is required, leaving the effect discussed here more robust. We quantitatively tested the backscattering of the wave under Bragg and locally-resonant scattering conditions with respect to edge roughness, where strongest backscattering is expected, and found that again, in the limit of $B/\mu \to 0$, the wave is indeed backscattering free (see Supplementary Materials for details).

## One-way transport in twisted kagome lattices

To demonstrate this spin-dependent edge transport experimentally, we choose the twisted kagome lattice, which is known to exhibit $B/\mu = 0$ in the ideal pin-joint limit (no bending stiffness between triangles)[40]. The lattice we consider herein (Fig. 4a) deviates from this limit, as we use beam-like elastic connectors instead of pin joints. We measure the Poisson's ratio of this lattice to be $\nu \simeq -0.488$ (see Supplementary Materials), corresponding to $B/\mu = 0.344$.

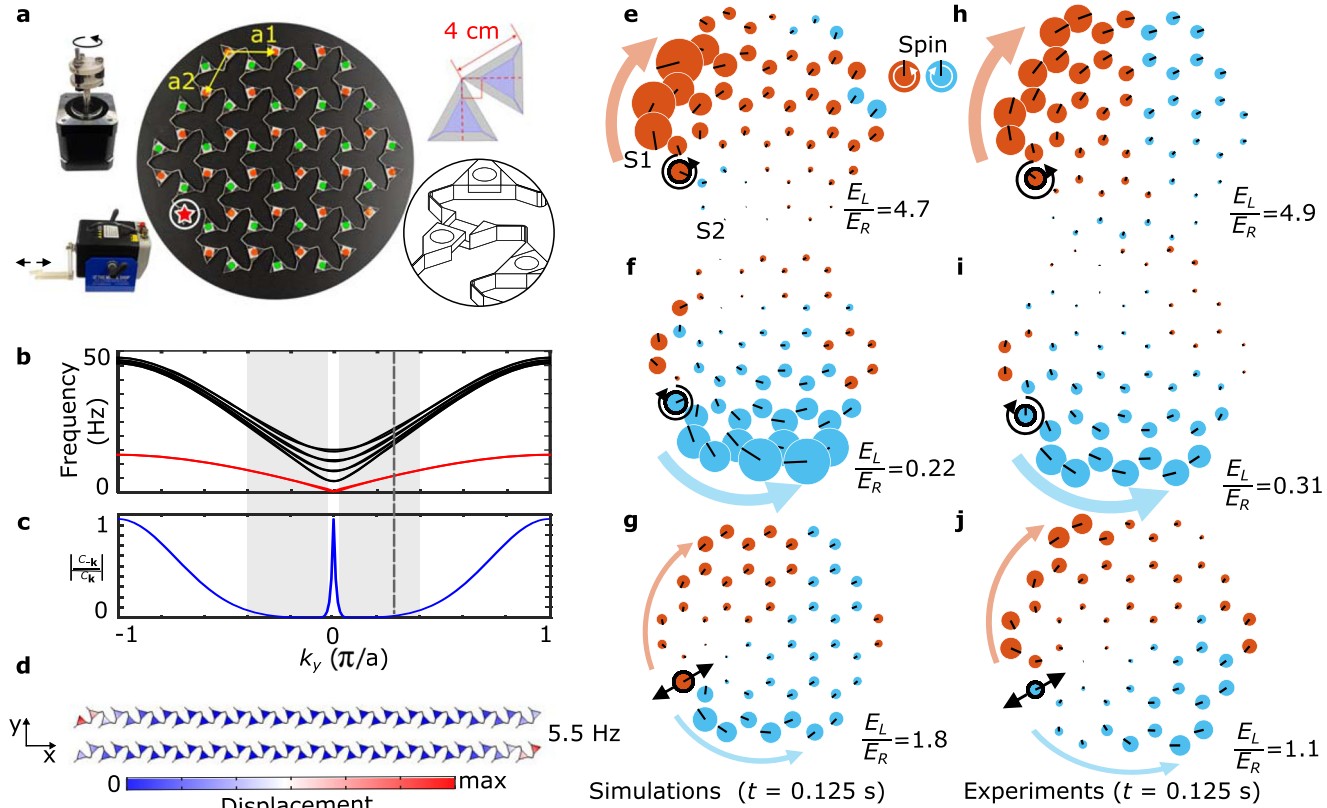

**Fig. 4 | Experimental verification of spin-momentum locking and one-way edge waves in an auxetic Maxwell lattice. a** The self-dual twisted kagome lattice[56]. The red star marks the actuation point. Red and green solid circles denote the triangles' centroids, which are used for image tracking. The 2D lattice has primitive basis vectors ($a_1$, $a_2$) with a lattice constant of $a = 5.71$ cm. Insets: (left) circular actuator (with a radius of 5 mm), and linear actuator; and (right) unit-cell geometry (gray: triangles with ideal pin-joint; blue: triangles with beam-like elastic connectors) and assembly diagram. **b** Super-cell band structures simulated in FEM for the elastic lattice in (**a**) with edge bands marked in red. **c** $|C_{-\mathbf{k}}/C_{\mathbf{k}}|$ of the lowest edge band. Shaded area indicates $|C_{-\mathbf{k}}/C_{\mathbf{k}}| < 0.1$. Dashed line marks $k_y = 0.28\pi/a$. **d** Two simulated edge mode profiles at $k_y = 0.28\pi/a$, also showing the geometry simulated in (**b**). **e**–**g** Time domain FEM simulations and **h**–**j** experimental results. Black circles and arrows indicate the location and the direction of the excitation, respectively. Solid circle sizes are proportional to velocity magnitudes, and black lines indicate velocity vector directions. Orange (light blue) color indicates the sign of the rotation of the triangle centroids' velocity vector, corresponding to counterclockwise (clockwise) spins. $E_L/E_R$ indicates the energy ratio between left (L) and right (R) propagating modes.

Pairs of branches of exponentially localized edge zero modes are known to arise in twisted kagome lattices under mixed (periodic horizontal and open vertical) boundary conditions, at opposite open edges[40]. They show up as the red branch in (Fig. 4b) via FEM super-cell analysis of the elastic kagome lattice, which are actually two nearly overlapping branches. We note that for this discrete lattice case, this branch of Rayleigh waves is not infinite (being subject to an upper cutoff). However, it still remains below all bands and extends to the $\omega = 0$ limit. At wavelengths approaching the width of the simulated domain (or small $\mathbf{k}$), the two localized modes generally hybridize into Lamb-like modes[44] (see Supplementary Materials). The effect of this hybridization can be seen in Fig. 4c, where $|C_{-\mathbf{k}}/C_{\mathbf{k}}| = 0$ exhibits a peak near $\mathbf{k} \to 0$. An important exception to this is the $B/\mu \to 0$ case, wherein the two edge modes are exactly orthogonal and, thus, do not hybridize and instead retain their asymmetric behavior. When $|\mathbf{k}| = 0.28\pi/a$, the decay length is short (about 1.4 unit cells,) and the hybridization is weak, so we choose the frequency of this pair of modes (5.5 Hz, Fig. 4d) for FEM and experiment of edge waves.

We verify spin-momentum locking and one-way edge wave propagation by actuating at the centroid of one triangle at the edge. With circularly polarized excitation, we observe spin-dependent one-way edge wave propagation in both simulation (Fig. 4e, f) and experiment (Fig. 4h, i). With linear actuation, we observe nearly symmetric edge-wave propagation, wherein the wave's spin is matched to the direction of propagation (Fig. 4g, j). For each excitation

case, we also calculate the energy ratio $E_L/E_R$ (Fig. 4e–j) between L and R propagating edge modes. The energy for each mode is calculated by integrating the average velocity squared (over $0.01 - 0.15$ s) of triangles on the edge to the left and right of the excitation point (see Supplementary Materials for specific triangles). Both the spatial response and the $E_L/E_R$ ratio show a close agreement between simulation and experiment (Fig. 4).

## Discussion

We have reported a distinct type of one-way and backscattering-free edge wave based on spin-momentum locking in 2D auxetic materials with moduli $B/\mu \to 0$. These modes are protected by a topological winding number that generalizes discrete angular momentum eigenvalues to arbitrary geometries. Unlike other topologically protected edge modes, these protected edge modes are not required to reside in gaps between bulk bands and instead exist for a broad range of frequencies. This is important, because, compared to the bandgap-tied mechanisms (quantum Hall, quantum spin, valley, or mirror Hall types) it: (i) opens infinite bandwidth and especially permits previously inaccessible ranges of low frequencies for topological transport, enabling access to material regimes with reduced dissipation, and (ii) does not necessitate any precise periodic or locally-resonant structures (simplifying fabrication).

As per its broadband functionality, the one-way edge mode transport mechanism shown herein is applicable to essentially both

auxetic lattices and continua in the $B/\mu \to 0$ limit. As auxetic materials have been studied for decades[45], there exists a broad potential set of candidate materials, including top-down manufactured structural materials, such as 3D printed re-entrant honeycomb lattices[46], auxetic origami, and kirigami metamaterials[47,48], and processed bottom-up manufactured materials, such as thermomechanically compressed polyurethane foams[49]. Some naturally occurring auxetic materials have also been found, exhibiting anisotropic negative Poisson's ratios (albeit still far from $v \to -1$), such as $\alpha$-cristobalite and some metals[50]. Our theory predicts that asymmetric edge transport exists in these materials, although perfect backscattering-free one-way transport calls for proximity to $v \to -1$ and low dissipation.

Finally, we emphasize that the "spin" of these edge waves are real, in the sense that they are directly connected to angular momentum, instead of the "pseudo spins" typically unitized to introduce the spin Hall effect in phononic systems[10,16,17]. Because of this physical spin, it has been previously suggested that such circularly polarized waves may be able to couple to spin degrees of freedom of other signals[24] such as photons[51–53] and magnons[54]. Similarly, we speculate that such coupling and topological protection may find use in optomechanical device strategies[55].

## Methods

### Maxwell lattice manufacturing

The self-dual kagome lattice with 90° rotated solid equilateral triangles connected via long and slender beams shown in Fig. 4a was designed to be assembled together from repeating sub-lattice elements. The sub-lattice elements (design shown in Supplementary Materials) were machined from 6.2 mm thick polycarbonate sheets via a desktop CNC mill and then assembled together via press fits.

### Experimental edge wave characterization

A misaligned shaft coupling (5 mm central offset) is installed on a stepper motor to induce a circular displacement path to the actuated triangle. The motor is controlled by a RAMBo board via MATLAB R2022b. A ball bearing is used at the end of the shaft to minimize rotation of the actuated triangle. The excitation frequency is 5.5 Hz. An electrodynamic exciter (The Modal Shop, Smartshaker, K2007E01) is driven sinusoidally at 5.5 Hz by a signal generator (Tektronix, AFG3022C) to perform linear actuation. A digital camera records the displacements of the lattice under actuation from above at a sampling rate of 240 Hz. The locations of the red and green markers in Fig. 4a are tracked from the recorded image sequence by using the *imfindcircles* command in MATLAB R2022b. The spatial measurement resolution is calculated to be 0.37 mm for the circular actuation cases, and 0.48 mm for the linear actuation cases. The location coordinates are smoothed in the time domain with a Gaussian-weighted moving average filter using the *smoothdata* command with a window size of 20 in MATLAB R2022b.

### FEM simulation

The FEM simulations are performed using the COMSOL Multiphysics Structural Mechanics Module. The 2D and 3D continua depicted in Figs. 1, 2, and 3 use user-defined material properties with a mass density of 960 kg/m³ and a shear modulus of 0.01 GPa, and differing bulk moduli, with extra fine mesh (Fig. 1: Mapped; Figs. 2, 3: Free Triangular, minimum element size 0.0075 cm and maximum element size 2 cm). The plane stress approximation has been employed in the 2D simulations. The dimensions of these systems are approximately 1 m × 1 m for 2D (1 m × 1 m × 1 m for 3D). Figures 1 and 2 are simulated in the time domain, with a time step of $T/10$ s ($T = 1/f$, where $f$ is the excitation frequency), and the simulations are carried out from 0 s to $30T$ s. Figure 3 is simulated in the frequency domain. The governing equation is given by $\rho \frac{d^2 \mathbf{u}}{dt^2} = \nabla \sigma + \mathbf{F}(\mathbf{r}, t)$,

where $\mathbf{F}(\mathbf{r}, t)$ represents the time-dependent excitation at a boundary point, expressed in the form of $\boldsymbol{\psi}_{\mathbf{k}}(\mathbf{r}, t)$ (noting that a prescribed displacement is applied). In the 2D continua, the excitation spin direction is out-of-plane, while in the 3D continua, it is parallel to the surface. In Fig. 1, a free boundary condition is applied to the boundary with excitation, and low-reflecting boundary conditions are employed at the remaining boundaries. This setup allows us to evaluate the ratio $|C_{-\mathbf{k}}/C_{\mathbf{k}}|$. This ratio is calculated by taking the ratio of the square root of the energy at the marked points, which are located on opposite sides of the excitation, over an extended period of time.

In Figs. 2 and 3, free boundary conditions are applied. The excitation frequency is adjusted to 1000 Hz for different $B_{3D}/\mu$ in 3D. The excitation frequency varies depending on the value of $B_{3D}/\mu$ in 2D, with frequencies of 500 Hz, 1500 Hz, and 2000 Hz used for $B_{3D}/\mu = 0.01$, $B_{3D}/\mu = 0.1$, and $B_{3D}/\mu = 1$, respectively. The simulated results are smoothed in the time domain following the same process as the experimental data.

The Maxwell lattices shown in Fig. 4 are filled with polycarbonate material, which has a mass density of 1190 kg/m³, Young's modulus of 0.751 GPa, and a Poisson's ratio of 0.3182. In the super-cell analysis, periodic boundary conditions are applied along the y-direction to conduct parametric sweeps of wave vectors. The ratio $|C_{-\mathbf{k}}/C_{\mathbf{k}}| = 0$ in Fig. 4c is computed using equation (1), utilizing the eigenvectors (displacement) $\boldsymbol{\psi}_{\mathbf{k}}(\mathbf{r})$ and $\boldsymbol{\psi}_{-\mathbf{k}}(\mathbf{r})$ obtained for the first band. Here, $\mathbf{r}$ represents the centroid of the first left triangle, as the energy of the first band localizes at the left edge when $\mathbf{k}$ is not in the small limit. In Fig. 4(e–g), the excitation is applied at a single triangle centroid on the boundary, with a frequency $f = \omega/(2\pi) = 5.5$ Hz. This frequency corresponds to the lowest edge mode frequency in the super-cell analysis when $k_y = \pm 0.28\pi/a$, where $a$ is the lattice constant.

## Data availability

The data generated in this study have been deposited in the Zenodo database [https://zenodo.org/records/12752976].

## Code availability

The code used in this study have been deposited in the Zenodo database [https://zenodo.org/records/12752976].

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

## Acknowledgements

W.C., N.C., K.S., and X.M. acknowledge support from the Office of Naval Research (MURI N00014-20-1-2479). K.Q., N.B., and X.M. acknowledge support from the US Army Research Office (Grant No. W911NF-20-2-0182). The authors also acknowledge the support from ICAM-I2CAM through the SLiM-Ex exchange award.

## Author contributions

W.C. performed the full-wave simulations, K.Q. performed the fabrication, and experimental characterization. N.C., X.M. and K.S. developed the mathematical framework. W.C. and K.Q. analyzed and interpreted the data. All authors wrote the manuscript. N.B., X.M. and K.S. supervised the project.

## Competing interests

The authors declare no competing interests.
