## [Transparent Peer Review file · Nature Communications]

Backscattering-free edge states below all bands in two-dimensional auxetic media

Corresponding Author: Dr Wenting Cheng

Version 0:

Reviewer comments:

Reviewer #1

(Remarks to the Author)

This manuscript explores Rayleigh waves in two-dimensional media with a vanishing bulk modulus, leading to a perfect unidirectional propagation. The authors also introduce a new topological winding number derived from real space to describe these topological modes. It is commendable that the article not only provides detailed theoretical derivations but also conducts experimental verification using materials with negative Poisson's ratios. The results presented in this article are both intriguing and innovative, enhancing people's understanding of elastic waves and topology. Therefore, I recommend the publication of this article.

I have several concerns to address:

1. The author depicted the backscattering under different B/μ ratios in Figure 2. From the time-domain plot, varying B/μ ratios result in significant differences in wave velocities. My question is: Do wavelengths also vary significantly? It is well known that for defects of the same size, waves with longer wavelengths are less likely to be affected by the defect. If in Figure 2, when B/A is small, the wavelength of Rayleigh waves is larger, then the reason for less reflection may be the longer wavelength rather than the smaller B/μ ratio. It is recommended that the author demonstrate the impact of B/μ under the same wavelength conditions, which will enhance the credibility of Figure 2.
2. The author demonstrated the significant differences between 2D and 3D Rayleigh waves and provided a detailed derivation process in the supplementary materials. The derivation processes for 2D and 3D are remarkably similar. The main differences I can observe lie in the differences in bulk modulus and boundary conditions. However, it is still challenging to intuitively grasp the physical picture leading to these differences. Could the author provide references related to 2D Rayleigh waves for further insight?
3. The topological number defined by Eq.7 increases with frequency. For a semi-infinite medium (such as the scenario depicted in Fig.1), does the winding number become infinite? The authors mentioned that the Kan-Lubensky topological winding number differs from Eq.7. Does this imply that there is no bulk-edge correspondence in the 2D shear less sheet?
4. Authors have included both elastic spin and phonon spin in their considerations. However, the discussions about the elastic spin and phonon spin miss an important reference [Chinese Phys. Lett. 39, 126301], wherein the relation between elastic spin and phonon spin has been demonstrated clearly, as well as the orbital angular momentum of elastic waves. Also, authors may be interested in experiments about spin-momentum locking in Rayleigh-Lamb modes [Nat. Commun. 12, 6954 (2021)]&[Phys. Rev. Lett. 131, 136102 (2023)].
5. In Figure 1, the polarization of the 3D Rayleigh wave flips with increasing depth, and the spin direction of the elastic wave also reverses (for example, elastic spin demonstrated in [Phys. Rev. Lett. 131, 136102 (2023)]). Just out of curiosity: beyond the constraint of B/m approaching zero, is it possible to adjust other material parameters to achieve perfect circular polarization on the surface for 3D Rayleigh waves, similar to 2D Rayleigh waves?

Reviewer #2

(Remarks to the Author)

Please find attached the comments.

Reviewer #3

(Remarks to the Author)

Reviewer's Opinion:

The manuscript presents a novel approach to achieving backscattering-free edge states in two-dimensional auxetic media using spin-momentum locking of Rayleigh waves. The research addresses a significant challenge in the field of wave propagation, offering a passive and linear solution that can operate across a broad range of frequencies. The proposed methodology and experimental validation are compelling, and the findings have the potential to open new avenues for applications in phononic devices.

Overall Recommendation:

The manuscript presents a significant advancement in the study of elastic wave propagation. It has the potential to make a substantial contribution to the field and should be considered for publication.

Reviewer #4

(Remarks to the Author)

Version 1:

Reviewer comments:

Reviewer #1

(Remarks to the Author)

The author has answered all my questions. I appreciate all efforts during this review process. There is only two minor issue left:

1) In the sentence "Under the condition of a fixed wave length equal to the defect size, the immunity to backscattering increases as $B/\mu \rightarrow 0$ (see SM Fig. S2)." in the main text, (see SM Fig. S2) should probably be Fig. S3 or Fig. S4.

2) The explanations and derivations of formulas from S72 to S80 in the supplementary material are presented in detail in reference [Chinese Phys. Lett. 39, 126301]. I suggest that the author should cite this reference appropriately when using these formulas, which can help readers understand the angular momentum of elastic waves in more detail.

Reviewer #2

(Remarks to the Author)

I thank the authors for having carefully answered to all my concerns.

Specifically, I appreciated the answer to the novelty issue, which allowed me to better understand the novelty of the work and the differences with respect to what done in ref. [25] of the previous version of the manuscript.

I carefully went through also through the comments of the other reviewers and, on the basis of their reports, I think I can change my position and endorse the publication of this manuscript in Nature Communications.

I. REPLY TO REFEREE 1

“This manuscript explores Rayleigh waves in two-dimensional media with a vanishing bulk modulus, leading to a perfect unidirectional propagation. The authors also introduce a new topological winding number derived from real space to describe these topological modes. It is commendable that the article not only provides detailed theoretical derivations but also conducts experimental verification using materials with negative Poisson’s ratios. The results presented in this article are both intriguing and innovative, enhancing people’s understanding of elastic waves and topology. Therefore, I recommend the publication of this article.”

We appreciate very much the positive comments from the referee.

“I have several concerns to address:”

“1. The author depicted the backscattering under different B/μ ratios in Figure 2. From the time-domain plot, varying B/μ ratios result in significant differences in wave velocities. My question is: Do wavelengths also vary significantly? It is well known that for defects of the same size, waves with longer wavelengths are less likely to be affected by the defect. If in Figure 2, when B/μ is small, the wavelength of Rayleigh waves is larger, then the reason for less reflection may be the longer wavelength rather than the smaller B/μ ratio. It is recommended that the author demonstrate the impact of B/μ under the same wavelength conditions, which will enhance the credibility of Figure 2.”

We thank the referee for this great question. The backscattering-free transport we reported does not require the wavelength to be long. The effect of fixed wavelength and changing B/μ was demonstrated in the original manuscript Supplementary Materials, however, we did not explicitly point this out (which we have addressed in the revised manuscript). Specifically, in the original Fig. S2, the excitation frequency was chosen to provide a wavelength of 4 cm for different B/μ ratios. The goal of this example was to demonstrate the robustness of backscattering-free, one-way transport of edge waves with respect to a Bragg grating with a periodicity of 2 cm. The result shows that with a fixed wavelength with respect to the scatterer size, the backscattering immunity increases as B/μ approaches zero.

We have highlighted this point in the revised manuscript, as follows: **“In addition, this backscattering-free one-way transport is independent of the wavelength. Under the condition of a fixed wavelength equal to the defect size, the immunity to backscattering increases as $B/\mu \rightarrow 0$ (see SM Fig. S2).”**

“2.The author demonstrated the significant differences between 2D and 3D Rayleigh waves and provided a detailed derivation process in the supplementary materials. The derivation processes for 2D and 3D are remarkably similar. The main differences I can observe lie in the differences in bulk modulus and boundary conditions. However, it is still challenging to intuitively grasp the physical picture leading to these differences. Could the author provide references related to 2D Rayleigh waves for further insight?”

We thank the referee for this great suggestion. In the revised Supplementary Materials (Section SVII), we have added the following publication, which we believe to be the first derivation of Rayleigh waves in 2D media.

[3] J. Oliver, F. Press, and M. Ewing, Two-dimensional model seismology, *Geophysics* **19**, 202 (1954).

In addition, we have revised Sections SI and SII in the Supplementary Materials to clarify the comparative derivations of the 2D and 3D Rayleigh waves.

“3.The topological number defined by Eq.7 increases with frequency. For a semi-infinite medium (such as the scenario depicted in Fig.1), does the winding number become infinite? The authors mentioned that the Kane-Lubensky topological winding number differs from Eq.7. Does this imply that there is no bulk-edge correspondence in the 2D shear less sheet?”

We thank the referee for this excellent question. For each mode in a semi-infinite medium, the winding number is infinite. As the referee pointed out, the conventional bulk-edge correspondence does not apply to the systems studied here. This is because, although both conformal symmetry and topological insulators can give rise to edge modes, these two phenomena are fundamentally different. Most importantly, in systems with conformal symmetry, the edge modes are not fully dictated by the bulk properties. Instead, the edge itself plays a significant role.

In our study, as shown in the derivation of conformal invariance of the zero mode at $B/\mu = 0$ (Section SV in the Supplementary Materials), the origin of conformal invariance depends on both the bulk equations and the boundary conditions. This dependence on both bulk and edge properties demonstrates that the bulk alone is insufficient to

determine the fate of the system, thereby invalidating the conventional bulk-edge correspondence.

The referee’s observation highlights a crucial point and one of the key results of this study: the backscattering-free edge transport obtained here represents a fundamentally new mechanism, distinct from those discussed in the context of topological insulators and bulk-edge correspondence. Remarkably, the winding number we defined provides a new type of topological protection to the linear momentum of the edge wave, as shown in Eq. (11).

We have revised the manuscript to better highlight this point (page 4, lower right), as follows: “...These two types of winding numbers describe completely different phenomena and they are not interchangeable. Kane-Lubensky winding numbers, along with other winding numbers based on momentum space winding of wave functions, describes how the bulk states govern the edge modes through the bulk-edge correspondence. In contrast, the winding number defined here is a winding in real space and is based on conformal symmetry in the $B/\mu \rightarrow 0$ limit, which comes from both the bulk equation-of-motion and the free boundary conditions. This phenomenon is fundamentally new and cannot be characterized by conventional winding numbers and bulk-edge correspondence.”

“4. Authors have included both elastic spin and phonon spin in their considerations. However, the discussions about the elastic spin and phonon spin miss an important reference [Chinese Phys. Lett. 39, 126301], wherein the relation between elastic spin and phonon spin has been demonstrated clearly, as well as the orbital angular momentum of elastic waves. Also, authors may be interested in experiments about spin-momentum locking in Rayleigh-Lamb modes [Nat. Commun. 12, 6954 (2021)]&[Phys. Rev. Lett. 131, 136102 (2023)].”

We appreciate the effort of the referee recommending these valuable references. In the revised main text, we have cited the following publications.

[29] W. Yuan, C. Yang, D. Zhang, Y. Long, Y. Pan, Z. Zhong, H. Chen, J. Zhao, and J. Ren, Observation of elastic spin with chiral meta-sources, Nature communications 12, 6954 (2021).

[30] C. Yang, D. Zhang, J. Zhao, W. Gao, W. Yuan, Y. Long, Y. Pan, H. Chen, F. Nori, K. Y. Bliokh, et al., Hybrid spin and anomalous spin-momentum locking in surface elastic waves, Physical Review Letters 131, 136102 (2023).

[31] J. Ren, From elastic spin to phonon spin: symmetry and fundamental relations, Chinese Physics Letters 39, 126301 (2022).

“5. In Figure 1, the polarization of the 3D Rayleigh wave flips with increasing depth, and the spin direction of the elastic wave also reverses (for example, elastic spin demonstrated in [Phys. Rev. Lett. 131, 136102 (2023)]). Just out of curiosity: beyond the constraint of B/μ approaching zero, is it possible to adjust other material parameters to achieve perfect circular polarization on the surface for 3D Rayleigh waves, similar to 2D Rayleigh waves?”

We thank the referee for this great question. For isotropic materials, we cannot achieve perfect circular polarization on the surface for 3D Rayleigh waves for any value of B_{3D}/μ . To achieve perfect circular polarization of surface waves, anisotropy is necessary. Mathematically, this question can be framed as follows: “Can a 3D elastic system with a generic elastic tensor C_{ijkl} give rise to fully circular polarization?” This question remains open and is of significant importance. At present, we do not have a definitive answer, but it is certainly a worthwhile subject for future investigation.

II. REPLY TO REFEREE 2

“The manuscript by Cheng et al. proposes a metamaterial design to achieve “unidirectional and backscatteringfree propagation in elastic media and lattices”. The claim of topological protection of these modes is made based on a “new defined topological winding number” that, according to what authors state “plays the role of discrete angular momentum eigenvalues despite the arbitrary geometry and protects the linear momentum of the wave along the edge”.

“The research methodology is based on analytical and numerical calculations supporting measurements.”

“As a general statement, that the topic falls within the very active research line of elastic topological insulators and I think that it could be of interest to the scientific community. However, the following critical points prevent me from endorsing this work for publication in a journal of the caliber of Nature Communications in the present form:”

We thank the referee for the effort and time they put in reviewing our manuscript. Below we respond to each question raised by the referee.

11. Novelty

“To the best of my understanding, one of the major claims of novelty of the paper is to design a metamaterial so that $B/\mu \rightarrow 0$, so that edge modes are orthogonal, thus, not hybridizing (and retaining their asymmetric behavior). This is a rather good idea. However, this has already been proposed in Ref. [25] of the current manuscript, where “sound waves waveguiding with suppressed backscattering when scatters fail to flip the acoustic spin” was reported and achieved via comb-like metasurfaces by “imposing a soft boundary of the π reflection phase”. Also, orthogonality of modes in twisted kagome lattices under mixed (periodic horizontal and open vertical) boundary conditions at opposite open edges was reported in Ref. [34] of the current manuscript. Consequently, it is not totally clear to me how the proposed research provides a leap forward with respect to the current state of the art, especially given the requirements of novelty of a journal of the caliber of Nat. Commun.”

We sincerely appreciate the referee’s valuable feedback, which highlights an important point: it is our responsibility to clearly present and sufficiently emphasize the novelty of our study, especially the progress we have made compared to existing works. Following the referee’s suggestions, we have revised the manuscript to better compare our study with prior research. This feedback has been immensely helpful in improving the quality and clarity of our work, and we are grateful for it.

As the referee noted, backscattering-free transport has been the focus of many previous studies. However, it is crucial to emphasize that the phenomena and underlying principles proposed in this study are fundamentally different from those in existing works, including Refs. [28] and [40] in the current manuscript (previously Refs. [25] and [34]). This new mechanism features remarkable robustness against any edge roughness, does not require any particular microstructure or symmetries, and thus opens new avenues for achieving backscattering-free one-way waves in practice.

In particular, the one-way transport described in Ref. [28] relies on symmetry protection and therefore lacks topological robustness. Perturbations that break the required symmetry will induce backscattering and disrupt the one-way transport. In contrast, our approach achieves topologically protected one-way transport that is entirely immune to edge roughness, eliminating the need for symmetry protection. This robustness against perturbations, regardless of their symmetry-breaking patterns, is a key distinction between our system and the one in Ref. [28]. The physical setting of our work is also significantly distinct from Ref. [28]. While Ref. [28] focuses on 1D phononic modes, our work introduces physics unique to 2D elastic media, offering a novel perspective and advancing the understanding of transport phenomena in this context.

In regards to Ref. [40], that work demonstrates the orthogonality of edge modes in twisted kagome lattices. This result of Ref. [40] alone does not ensure topological protection against backscattering, nor does it connect orthogonality to the intrinsic dynamical properties of the system. Furthermore, it is limited to discrete lattices. In contrast, our paper provides a general framework for achieving topological protection against backscattering for orthogonal edge modes, which can be broadly applied to both continuum and discrete 2D elastic media.

We have revised the manuscript to clarify these points in the introduction, as follows: *“...It is important to note that the spin here is the true physical spin of the elastic waves in continuous media, and the underlying principles proposed in this study, based on conformal symmetry, are fundamentally different from those in existing literature. This new mechanism exhibits exceptional robustness against edge roughness, does not require specific microstructures or symmetries, and thus opens new pathways for achieving backscattering-free one-way waves in practice. We experimentally verify this effect using auxetic Maxwell lattices (which enable $B/\mu \rightarrow 0$). We formulate an analytic theory to show that this locking is topologically protected by a new type of real space winding number of the wave function and thus robust against any edge roughness, endowing true “backscattering-free” one-way transport to these waves, and demonstrate this effect computationally. Because of the aforementioned infinite bandwidth, wherein our modes exist below all bulk bands (in the sense of having lower wavespeed), we also highlight that these modes can occur at arbitrarily low frequencies.”*

12. Topological protection

“Authors define the wave propagation as topologically protected by defining a “new topological winding number that protects the linear momentum of the wave along the edge”. However, it is not clear how this relates to topological protection. It would be very useful if authors provide more insight on why they used this alternative approach. In addition, I am not sure that the proposed definition could “properly” characterize the “topological” nature of the

observed modes. Would it be possible to apply this method to “ordinary topologically protected modes” to see what they would give. In conclusion, I would suggest rephrasing these parts or providing additional justification for this.”

We appreciate the referee for raising this interesting question and for their useful suggestions. To address the referee’s question, in short, the new topological winding number we propose and the traditional ones apply to two types of completely different edge states. The former is concerned with edge waves below all bulk bands, which originates from conformal symmetry, and the latter is concerned with band gap edge waves originating from the topology of the bulk bands.

Traditional topological winding numbers are defined in k -space, where the integration of the eigenvectors of the bands above and below the band gap yields a topological invariant. If the topological invariant is non-zero, the system is topologically non-trivial, and, according to the bulk-edge correspondence principle, the traditional topological winding number guarantees the existence of edge states.

However, in our work, we investigate free boundary systems, specifically focusing on edge state bands (such as Rayleigh waves in continua and floppy modes in lattices) that lie below all bulk bands and lack a band gap, making traditional topological winding numbers and the bulk-edge correspondence principle inapplicable. Therefore, we define a new topological winding number, which is defined in real space. In Page 4, lower right, we had emphasized that this form of topological protection is different from traditional topologically protected modes, as our topological winding number protects one-way transport. Additionally, in Page 5, Column 1, Paragraph 2, we had provided a detailed explanation of how this new topological winding number protects the linear momentum of the wave along the edge, ensuring the one-way propagation.

In the revised manuscript, we have further clarified these points (Page 4, lower right), as follows: “...These two types of winding numbers describe completely different phenomena and they are not interchangeable. Kane-Lubensky winding numbers, along with other winding numbers based on momentum space winding of wave functions, describes how the bulk states govern the edge modes through the bulk-edge correspondence. In contrast, the winding number defined here is a winding in real space and is based on conformal symmetry in the $B/\mu \rightarrow 0$ limit, which comes from both the bulk equation-of-motion and the free boundary conditions. This phenomenon is fundamentally new and cannot be characterized by conventional winding numbers and bulk-edge correspondence.”

I3. Additional comments

“a. Results provided in Fig. 2 are very interesting. However, it seems that the propagation can only be achieved along the external boundary of the domain. Would it be possible to achieve the propagation also inside the domain, as usually shown in “ordinary” topologically protected media? How would the interface be designed in this case?”

This is an important question, and we have revised our manuscript to clarify this point.

The mechanism of one-way propagation reported in our study cannot be achieved at an interface; rather, it is a free-boundary phenomenon. This represents a key distinction between our work and existing studies based on topological states, such as topological insulators. As discussed, while our phenomenon also involves a topological winding number, it is defined in real space with open boundaries and is fundamentally different from the k -space-based topological indices of topological insulators and other related topological states. Consequently, the edge states in our system exhibit some very intriguing features. However, this difference from topological insulators also implies that certain properties expected in k -space-based topological insulators, such as domain wall modes, do not arise in our setup.

“b. Why are time steps displayed at so different instants? To make the comparison more direct, I would propose to normalize it (I guess as a function of the mechanical properties?).”

We thank the referee for this suggestion. In the revised manuscript Fig. 2(b), we now normalize the time t by $t_0 = L/c_R$, where L is the distance between points A and B , and $c_R = \xi(B/\mu)\sqrt{\mu/\rho}$ is the Rayleigh wave speed in 2D with $\xi(B/\mu)$ satisfying the equation

$$\xi(B/\mu)^6 - 8\xi(B/\mu)^4 + \left(8 + \frac{16B/\mu}{1+B/\mu}\right)\xi(B/\mu)^2 - \frac{16B/\mu}{1+B/\mu} = 0. \quad (1)$$

For the energy E in Fig. 2(b), we now also normalize it by $E_0 = \rho c_R^2 = \mu\xi(B/\mu)^2$.

“c. Do materials with $B/\mu \rightarrow 0.001$ withstand their own weight? What would be its mechanical resistance?”

This is a great question. Since it is only the ratio between B and μ that matters, we can make μ large so that B is not small, and the material can withstand its own weight. Second, as shown in the experiment, we don't need B/μ to be as small as 0.001 to achieve one-way transport. As shown in the Supplementary Materials, the experimental setup corresponds to $B_{3D}/\mu = 0.173$, and for this discrete system, we can already see the one-way transport behavior.

“d. Authors put the accent on “expanding the range of usable frequencies in each material” in opposition to “bandgap-tied mechanisms in the case of the analogues of quantum Hall, quantum spin, and valley effects”. However, these localized modes can span very large frequency ranges. The fact of observing them inside a BG is for observation purposes, which should be true also in this approach. In other words, how would authors measure the mode in a region where a total BG is not present?”

We thank the referee for raising this good question. For the analogues of quantum Hall, quantum spin Hall, and quantum Valley Hall effects, the requirement of a band gap implies that these effects cannot be achieved in the low frequency ($\omega \rightarrow 0$) regime (*i.e.*, below the lowest bulk acoustic band). Furthermore, unlike the aforementioned Hall effects, our one-way propagation phenomenon is highly broadband, as it is not limited to the band gap region. Indeed, for the edge wave in a semi-infinite medium, the bandwidth is theoretically infinite. In the revised Supplementary Materials, we have added a section (Section SVIII), as follows, to highlight this broadband feature: **“Unlike other topologically protected edge modes, the edge modes in this study are not confined to gaps between bulk bands. Instead, for our case of a 2D continuum half space, the edge modes exist for an infinitely broad range of frequencies. In Fig. S2, we demonstrate a one-way edge wave excited by a broadband clockwise Gaussian pulse. The Fourier transform of the excitation reveals a broad frequency range with a bandwidth of 200 Hz, as shown in Fig. S2(a). Using a semi-infinite 2D plane with $B_{3D}/\mu = 0.001$, the Gaussian pulse excitation was applied at a point on the bottom edge (Fig. S2(b)). Time-domain simulations show that the energy propagates to the right. The energy density as a function of time at points A and B further confirms the one-way nature of the edge mode.”**

Second, as regards observability, we note that, in any setting, the presence of a band gap is not needed for observation purposes. Rather, the presence of the mode in the band gap is a consequence of the requisite symmetry to enable the topologically protected edge modes. In any case, we directly demonstrate the observability of the modes via simulation and experimental measurement. The one-way nature is quantified by measuring the magnitude of the energy propagating to either side of the excitation, and the backscattering-free nature is measured by measuring the reflected and transmitted wave amplitudes after the one-way mode has already been formed.

“e. It's not clear why authors use different types of formalisms: Rayleigh waves, 2D continua, lattices. I found this a little bit confusing. Can authors please clarify this point?”

We attempt to clarify the need for our use of different formalisms, as follows. We first contrast our findings against Rayleigh waves, which traditionally exist in 3D continuous media. However, as shown in the manuscript, 3D Rayleigh waves do not support the one-way transport and backscattering-free property. Our contribution in this manuscript is that we show that such a property exists for Rayleigh wave analogs in 2D continua. We then introduce the lattice, as it is a convenient mechanism to obtain small B/μ (or, equivalently, Poisson's ratio approaching -1) in practice for our experimental observation of this phenomenon.

“f. May I also please ask the authors to add a second level of reading? I found the accessibility to the nonspecialist reader of this section is very limited.”

We have edited throughout the main text and the supplementary information to improve the readability, and hoped that we have effectively increased the accessibility of our manuscript.

“g. The context of topologically protection in elasticity is not thoroughly described. I would consider citing the following review papers: i) 2019, Nature Reviews Physics 1 (4), 281-294 – Topological phases in acoustic and mechanical systems. ii) 2020, Comptes Rendus Physique 21.4-5 (2020): 467-499 – Topological wave insulators: a review. iii) 2021, Journal of Applied Physics 130 (14) – Design of topological elastic waveguides”.

We appreciate the effort of the referee recommending these valuable references. In the revised manuscript, we have cited the following publications.

- [11] F. Zangeneh-Nejad, A. Alu, and R. Fleury, Topological wave insulators: a review, Comptes Rendus.

Physique 21, 467 (2020).

[12] G. Ma, M. Xiao, and C. T. Chan, Topological phases in acoustic and mechanical systems, Nature Reviews Physics 1, 281 (2019).

[13] M. Miniaci and R. Pal, Design of topological elastic waveguides, Journal of Applied Physics 130 (2021).

III. REPLY TO REFEREE 3

Reviewer's Opinion:

The manuscript presents a novel approach to achieving backscattering-free edge states in two-dimensional auxetic media using spin-momentum locking of Rayleigh waves. The research addresses a significant challenge in the field of wave propagation, offering a passive and linear solution that can operate across a broad range of frequencies. The proposed methodology and experimental validation are compelling, and the findings have the potential to open new avenues for applications in phononic devices.

Overall Recommendation:

The manuscript presents a significant advancement in the study of elastic wave propagation. It has the potential to make a substantial contribution to the field and should be considered for publication.

We appreciate very much the positive comments from the referee.

IV. REPLY TO REFEREE 4

We thank the referee for the effort and time they put in reviewing our manuscript.

point by point.

Reviewer 1:

Comment: *In the sentence “Under the condition of a fixed wave length equal to the defect size, the immunity to backscattering increases as $B/\mu \rightarrow 0$ (see SM Fig. S2).” in the main text, (see SM Fig. S2) should probably be Fig. S3 or Fig. S4.*

Response: We thank the reviewer for pointing this out. The correct reference should indeed be to Fig. S3 and Fig. S4, and we have updated the text accordingly.

Comment: *The explanations and derivations of formulas from S72 to S80 in the supplementary material are presented in detail in reference [Chinese Phys. Lett. 39, 126301]. I suggest that the author should cite this reference appropriately when using these formulas, which can help readers understand the angular momentum of elastic waves in more detail.*

Response: We appreciate the reviewer’s suggestion. We have added a citation to the reference [Chinese Phys. Lett. 39, 126301] in the relevant sections of the supplementary material (S72 to S80) to provide readers with additional context regarding the angular momentum of elastic waves.

Reviewer 2:

Comment: *I thank the authors for having carefully answered to all my concerns. Specifically, I appreciated the answer to the novelty issue, which allowed me to better understand the novelty of the work and the differences with respect to what done in ref. [25] of the previous version of the manuscript. I carefully went through also through the comments of the other reviewers and, on the basis of their reports, I think I can change my position and endorse the publication of this manuscript in Nature Communications.*

Response: We thank the reviewer for their comments and for endorsing the publication of our manuscript.

Here is a brief summary of the main findings of the paper, as requested in the author checklist:

“We propose passive, linear edge states based on spin-momentum locking of Rayleigh waves in 2D media in the limit of zero bulk to shear modulus ratio, that give backscattering-free propagation immune to edge roughness with no frequency limitation.”

We hope these revisions meet the approval of the reviewers and editors. All required items have been submitted online. Thank you once again for accepting our work for publication in Nature Communications.

Best regards,

Wenting Cheng
 Kai Qian
 Nan Cheng
 Nicholas Boechler
 Xiaoming Mao
 Kai Sun

The manuscript by Cheng et al. proposes a metamaterial design to achieve “*unidirectional and backscattering-free propagation in elastic media and lattices*”. The claim of topological protection of these modes is made based on a “*new defined topological winding number*” that, according to what authors state “*plays the role of discrete angular momentum eigenvalues despite the arbitrary geometry and protects the linear momentum of the wave along the edge*”.

The research methodology is based on analytical and numerical calculations supporting measurements.

As a general statement, that the topic falls within the very active research line of elastic topological insulators and I think that it could be of interest to the scientific community. However, the following critical points prevent me from endorsing this work for publication in a journal of the caliber of Nature Communications in the present form:

I1. Novelty

To the best of my understanding, one of the major claims of novelty of the paper is to design a metamaterial so that $B/\mu \rightarrow 0$, so that edge modes are orthogonal, thus, not hybridizing (and retaining their asymmetric behavior). This is a rather good idea. However, this has already been proposed in Ref. [25] of the current manuscript, where “sound waves waveguiding with suppressed backscattering when scatters fail to flip the acoustic spin” was reported and achieved via comb-like metasurfaces by “imposing a soft boundary of the π reflection phase”.

Also, orthogonality of modes in twisted kagome lattices under mixed (periodic horizontal and open vertical) boundary conditions at opposite open edges was reported in Ref. [34] of the current manuscript.

Consequently, it is not totally clear to me how the proposed research provides a leap forward with respect to the current state of the art, especially given the requirements of novelty of a journal of the caliber of Nat Commun.

I2. Topological protection

Authors define the wave propagation as topologically protected by defining a “new topological winding number that protects the linear momentum of the wave along the edge”. However, it is not clear how this relates to topological protection. It would be very useful if authors provide more insight on why they used this alternative approach.

In addition, I am not sure that the proposed definition could “properly” characterize the “topological” nature of the observed modes. Would it be possible to apply this method to “ordinary topologically protected modes” to see what they would give.

In conclusion, I would suggest rephrasing these parts or providing additional justification for this.

I3. Additional comments

a. Results provided in Fig. 2 are very interesting. However, it seems that the propagation can only be achieved along the external boundary of the domain. Would it be possible to achieve the propagation also inside the domain, as usually shown in “ordinary” topologically protected media? How would the interface be designed in this case?

b. Why are time steps displayed at so different instants? To make the comparison more direct, I would propose to normalize it (I guess as a function of the mechanical properties?).

c. Do materials with $B/\mu \rightarrow 0.001$ withstand their own weight? What would be its mechanical resistance?

d. Authors put the accent on “expanding the range of usable frequencies in each material” in opposition to “bandgap-tied mechanisms in the case of the analogues of quantum Hall, quantum spin, and valley effects”. However, these localized modes can span very large frequency ranges. The fact of observing them inside a BG is for observation purposes, which should be true also in this approach. In other words, how would authors measure the mode in a region where a total BG is not present?

e. It’s not clear why authors use different types of formalisms: Rayleigh waves, 2D continua, lattices. I found this a little bit confusing. Can authors please clarify this point?

f. May I also please ask the authors to add a second level of reading? I found the accessibility to the non-specialist reader of this section is very limited.

g. The context of topologically protection in elasticity is not thoroughly described. I would consider citing the following review papers:

i) 2019, Nature Reviews Physics 1 (4), 281-294 – Topological phases in acoustic and mechanical systems

ii) 2020, Comptes Rendus Physique 21.4-5 (2020): 467-499 – Topological wave insulators: a review

iii) 2021, Journal of Applied Physics 130 (14) – Design of topological elastic waveguides